# Vestibular Versus Cochlear Stimulation on the Relief of Phantom Pain After Traumatic Finger Amputation

**DOI:** 10.3390/biomedicines13071601

**Published:** 2025-06-30

**Authors:** José Joaquín Díaz-López, José Adán Miguel-Puga, María Isabel Jaime-Esquivias, Maricela Peña-Chávez, Kathrine Jáuregui-Renaud

**Affiliations:** 1UMAE Hospital de Traumatología y Ortopedia Lomas Verdes, Instituto Mexicano del Seguro Social, Naucalpan 53150, Mexico; jose.diazlo@imss.gob.mx; 2Unidad de Investigación Médica en Otoneurología, Instituto Mexicano del Seguro Social, Ciudad de Mexico 06720, Mexico; adan.miguel@imss.gob.mx; 3Unidad de Medicina Física y de Rehabilitación Siglo XXI, Instituto Mexicano del Seguro Social, Ciudad de Mexico 04980, Mexico; maria.jaimee@imss.gob.mx (M.I.J.-E.); maricela.penac@imss.gob.mx (M.P.-C.)

**Keywords:** vestibular, phantom pain, depersonalization/derealization, traumatic amputation, hand injury

## Abstract

**Objective:** The aim of this study was to assess the effects of vestibular stimulation (semicircular canals/utricles) compared to cochlear stimulation on phantom pain and depersonalization/derealization symptoms after ≥3 months since traumatic amputation of hand-finger(s). **Methods**: A total of 125 adults (38.2 ± 8.1 years old) with phantom pain after amputation of one to four fingers agreed to participate. None of them wore prosthetic devices or had history of otology/audiology/vestibular/neurology/rheumatology/orthopedic/psychiatry disorders or psychopharmacological treatment. After a preliminary assessment, in a random order, they were exposed to caloric stimulation (right/left 44 °C/30 °C), centrifuge (right/left), and transient evoked otoacoustic emissions (TOAEs, right/left) with a follow-up of three days in between. Immediately before and after each stimulus, they reported on their pain characteristics and depersonalization/derealization symptoms. **Results**: After vestibular stimulation, a decrease in pain intensity was reported by at least one-third of the participants, which persisted for at least one day in the majority of them. Less than one-sixth of the participants reported pain decrease after cochlear stimulation. No influence was observed based on the side of the stimulation or the temperature, but the stimuli sequence had an effect. The centrifuge and TOAE effects were related to anxiety/depression symptoms and mainly observed when they were the first stimulus used. After caloric stimulation, pain decrease was independent from the sequence of the stimuli, and it was related to reports of feeling estrangement from the body. **Conclusions**: Mild caloric vestibular stimulation, whether applied to the right or left side and using warm or cold temperature, can modulate phantom pain after amputation of hand-finger(s) in patients with altered bodily sensations. However, individual cofactors may influence one’s susceptibility to experiencing this effect.

## 1. Introduction

Phantom pain refers to the pain perceived in a part of the body that is no longer present; this experience occurs after about 80% of amputations [1]. It is a complex phenomenon that can be severe, and it is difficult to treat (for review, see [2]). This distorted perception can persist for years [3], usually through intermittent pain episodes [4], with decreased health-related quality of life [5].

Motor and sensory remodeling of both the peripheral and the central nervous system occur after amputation, depending on the individual context (for review, see [6]). Plasticity produces a cascade of cortical reorganization at a network-level scale [7]. Relief of phantom pain by noninvasive transcranial stimulation can be predicted by the activity in pain-related brain areas, including the mid, posterior, and rostro-dorsal posterior insula [8,9].

Additionally, chronic pain can be related to deterioration of the body image [10]. In adults with limb amputation, those with phantom pain may report more negative body image perception and hypervigilance of the phantom limb compared to those with no painful phantom sensations [11].

The peripheral vestibular apparatus comprises five end-organs in each inner ear: three semicircular canals encoding angular acceleration, and two otolith maculae (utricle and saccule) encoding linear acceleration. Movements of the head, either active or passive, modify their spontaneous discharge (for review, see [12]). During natural movements, head rotation/tilt simultaneously stimulate both right and left end-organs (physiological stimuli) [13]. Conversely, asymmetric caloric stimuli (not physiological) can modify the spontaneous discharge of each horizontal semicircular canal by introducing either warm (excitatory) or cold (inhibitory) water/air into the corresponding external ear canal [14]. Likewise, stimulation of each utricle without tilt can be performed by unilateral centrifuge on a rotating chair that is shifted toward the right or the left of the rotation axis to produce a centripetally displaced g-force [15]. In addition, galvanic stimulation can have polarization effects on each vestibular nerve [16].

The vestibular afferents are continuously updated in a widespread central network, with influence on neurocognitive processes and perception (for review, see [17]). Evidence supports that vestibular stimulation may modulate both experimental and clinical pain [18,19,20,21,22]. In healthy volunteers, cold/left caloric vestibular stimulation can elicit modulation of both nociceptive processing and pain perception [20]. Galvanic vestibular stimulation has analgesic effects on experimental pain, accompanied by amplitude reductions of the vertex component of brain-evoked potentials [22]. In adults with a variety of central pain conditions, caloric stimulation with iced water can decrease pain intensity [19]. In patients with supracondylar amputation related to type 2 diabetes, either caloric vestibular stimulation (at 30°/44°) or unilateral centrifuge (300°/s peak, 3.85 cm) can modify the intensity of phantom limb pain while changing the perceptions of the self and the environment (depersonalization/derealization, DD) [21].

Evidence supports that vestibular stimulation may have general effects on the neural mechanisms underpinning the representation of the body [18]. Neuroimaging studies have shown that the main regions activated by caloric vestibular stimuli are located in the Sylvian fissure, insula, retro-insular cortex, fronto-parietal operculum, superior temporal gyrus, and cingulate cortex [23], including cortico-subcortical networks involved in cross-modal interactions between vestibular and somatosensory signals [24].

Intracranial recordings have shown that nociceptive inputs are processed in the insula in a posterior–anterior direction [25], with evidence on the relation of the posterior subdivision to somatosensory functions and pain perception [24]. The sensory mismatch induced by a non-physiological vestibular stimulus, with interaction in the posterior insular nociceptive networks, could contribute to the reduction in pain perception while updating the immediate experience of the body [21].

Before considering the potential application of vestibular stimulation in clinical practice, further research is required on the effects of various vestibular stimuli on phantom pain compared to control stimulation. To replicate and extend previous findings, this cross-over study was designed to assess the effects of caloric semicircular canal stimuli (right/left, cold/warm) and centrifuge utricle stimuli (right/left), compared to cochlear stimuli (right/left) by transient evoked otoacoustic emissions (TOAE) [26], on the intensity of phantom pain and DD symptoms reported by adults with persistent phantom pain after ≥3 months since traumatic amputation of hand-finger(s).

## 2. Materials and Methods

The study was conducted in accordance with the Declaration of Helsinki and its amendments and approved by the Institutional Review Board and Ethics Committee of Instituto Mexicano del Seguro Social (R-2018-785-071, date 25 July 2018). Informed consent was obtained from all subjects involved in the study.

### 2.1. Participants

A total of 125 volunteers, 24 women and 101 men, participated in the study (aged 38.2 ± 8.1 years; mean ± standard deviation); 107 were right-handed, four were left-handed, and 14 were ambidextrous (Edinburgh Inventory [27]). All of them reported phantom pain ≥ 3 months (median of 4 months, Quartile 1–Quartile 3 4 to 6 months) after traumatic amputation of one to four fingers (mode 1), 70 (56%, 95% confidence interval 47.2–64.7%) of the right hand, 54 (43.2%, 95% C.I. 34.5–51.8%) of the left hand, and one of both hands. Amputation of finger(s) of the dominant hand occurred in 63 (50.4%, 95% C.I. 41.6–59.1%) participants. Tobacco use was reported by 41 participants (32.8%, 95% C.I. 24.5–41.0%), alcohol consumption by 52 participants (41.6%, 95% C.I. 32.9–50.2%) (none with heavy use), and corrected refraction errors by 44 participants (35.2%, 95% C.I. 26.8–43.5%). The mean sleep time was 49.0 ± 7.4 h/week.

According to clinical records and direct interviews, none of the participants wore prosthetic devices or had history of otology/audiology/vestibular/neurology/rheumatology/orthopedic/psychiatry disorders or psychopharmacological treatment. However, 14 participants were receiving paracetamol, 2 were receiving diclofenac, 4 were receiving medical treatment for systemic high blood pressure (3 losartan/1 enalapril), and 1 was being treated for gastritis (omeprazole).

In addition, five more volunteers agreed to participate in the study but were unable to complete the study protocol due to personal reasons unrelated to the study or the phantom pain. This group included two women and three men, 21 to 46 years old.

The sample size was calculated for a cross-over design, for a 0.5 effect size of any vestibular stimuli, with a maximal difference of 0.2, power of 0.9, and a bilateral alpha of 0.05. At least 20 participants were included in each cluster.

### 2.2. Procedures

#### 2.2.1. Preliminary Evaluations

The diagnosis of phantom limb pain was confirmed by an independent surgeon. The general health and personal habits were documented by an in-house questionnaire, followed by a direct interview. The preliminary assessments comprised otology, audiology and vestibular evaluations. The vestibular evaluation included a questionnaire of symptoms related to balance [28] oculomotor recordings with rotational tests and the static subjective visual vertical (average of 10 estimations) (I-Portal NOTC, Neuro Kinetics, Pittsburgh, PA, USA).

#### 2.2.2. Questionnaires

Before the preliminary evaluations, the following instruments were administered for self-report:The *Douleur Neuropathique* 4 Questions (DN4) questionnaire [29], which includes 10 items: 7 on the quality of pain (burning, painful cold, and electric shocks) and abnormal sensations (tingling, pins and needles, numbness, and itching), and 3 on neurological signs in the painful area (touch hypoesthesia, pinprick hypoesthesia, and tactile allodynia). This instrument has 83% sensibility and 90% specificity for the diagnosis of neuropathic pain [29] and an intraclass correlation of 0.71 [30].The Lattinen Index [31] was used to assess chronic pain, including pain intensity and frequency, use of analgesics, functional ability, and sleep time, which are rated on a 5-point scale (from 0 to 4). This instrument has a Cronbach’s alpha of >0.7 and an intra-class correlation of >0.85 [31].A visual analog scale (VAS) was used to mark pain intensity as a point on a 100 mm horizontal line, representing “no pain” at the left end and the “most severe pain imaginable” at the right end, with an intraclass correlation > 0.8 [32]. At the first evaluation of the study, the repeatability coefficient was estimated as 11.8 mm [33], which is similar to the minimum change of 13 mm considered as clinically significant in patients with acute pain [34]. Therefore, in this study, a clinically significant change was considered when >12 mm.The Hospital Anxiety and Depression Scale (HADS) [35] comprises 14 items, including 7 for anxiety and 7 for depression, which are rated on a 4-point scale (from 0 to 3). Subscores were obtained by summing the ratings for the seven items of each subscale (range of 0–21), and a total score was obtained by summing the ratings of all items (range of 0–42), with Cronbach’s alpha ranging from 0.67 to 0.93 [36].The Dissociative Experiences Scale [37], which comprises 28 items, was used to assess disturbances in memory, identity, cognition, absorption, imaginative involvement, and feelings of derealization and depersonalization. Each item score ranges from 0%, “This never happens to you”, to 100%, “This always happens to you”, in multiples of ten. A total score was calculated by dividing the sum of the individual scores by 28 (range of 0 to 100%), with a mean Cronbach’s alpha of 0.93 and test–retest reliability of 0.78–0.93 [38].The Depersonalization/Derealization Inventory [39] that was designed to assess depersonalization/derealization (DD) symptoms in clinically anxiety states, enabling correlation with concurrent neurophysiological variables. It comprises 28 items rated on a 5-point scale (from 0 to 4). A total score was obtained by summing up the ratings of all items (range of 0–112), with a Cronbach’s alpha of 0.95 [39].

#### 2.2.3. Stimuli

The following three types of stimuli were administered by the same investigator:Caloric stimuli at 30 °C or at 44 °C (ICS NCI 480, Otometrics, Taastrup, Copenhagen, Denmark) was applied to either the right or the left ear, without eye-recording devices [13,14].Unilateral centrifuge (I-Portal NOTC, Neuro Kinetics, Pittsburgh, PA, USA) was applied either to the right or to the left (3.85 cm at 300°/s peak velocity). The chair accelerated to 300°/s in 60 s. After 60 s at full-speed rotation, it moved from the center position to the right/left over 30 s. It remained in the offset position for 60 s, then moved from the right/left position to the center over 30 s. Finally, it decelerated from 300°/s to 0°/s in 60 s.Transient evoked otoacoustic emissions (TOAEs) (1.5 to 4 kHz, 64 s averaging, clicks at 83 dB SPL peak equivalent) (OtoRead, Interacoustics, Assens, Denmark) were delivered monoaurally to either the right or the left ear.

Adequate stimulation was verified by the expected responses to each stimulus: duration of horizontal nystagmus and vertigo after caloric stimuli, deviation of the visual vertical during centrifuge, and at least three valid cochlear responses to TOAEs.

After the preliminary evaluations, the participants were randomly assigned exposure to a first stimulus, and then to a different stimulus per session, with three days in between, until they were exposed to the three types of stimuli. The actual sequences of the three stimuli were (1) caloric/centrifuge/TOAE; (2) caloric/TOAE/centrifuge; (3) centrifuge/caloric/TOAE; (4) centrifuge/TOAE/caloric; (5) TOAE/caloric/centrifuge; (6) TOAE/centrifuge/caloric.

The first stimulus was caloric in 42 participants (21 right/21 left), centrifuge in 41 participants (20 right/21 left), and TOAE in 42 participants (21 right/21 left). Since caloric stimuli can be either warm or cold, it was right/warm in 32 participants, right/cold in 32 participants, left/warm in 31 participants, and left/cold in 30 participants. Meanwhile, 60 participants were exposed to centrifuge stimulation on the right side and 65 on the left. TOAEs were evoked in the right ear in 64 participants and in the left ear in 61 participants.

#### 2.2.4. Pain Assessment

Immediately before and after each stimulus was applied, the participants reported on their pain characteristics using the DN4 [29] and pain intensity using the visual analog scale (VAS), which was included in both the DN4 (after item 3) and the Lattinen Index (at the end of the questionnaire). For analysis, the average of these two estimations was calculated.

During the week before the first stimulus, the three days in between each stimulus, and the three days after the third stimulus, the participants were instructed to report on any pain (printed dairy) using the VAS, and on accompanying sensations using the DN4 (items 4 to 8). They received daily standardized remainders by text message.

### 2.3. Analysis

Statistical analysis was performed on coded data to assess the immediate responses to the stimuli and the contribution of cofactors to the variance of the responses to each stimulus and among the stimuli. The analysis was performed using the “*t*” test, analysis of variance, and repeated measures multivariate analysis of covariance (MANCoVA) (CSS, Statsoft, Tulsa, OK, USA), with a two-sided significance level of 0.05.

## 3. Results

### 3.1. Bivariate Analysis of Pain Intensity

The pain characteristics and mental symptoms before any stimulation are described in Table 1. Moderate pain intensity prevailed, with burning and/or painful cold perceptions, associated with “tingling”, “pins and needles” and “numbness” sensations in more than half of the participants, and “itching” sensation in about 45% of them. The participants also reported mild to moderate common mental symptoms. Before the first stimulation, the pain characteristics and mental symptoms were similar among the randomized subgroups receiving any stimulus for the first time (Table 1).

Before any stimulation, the pain ratings were similar among the subgroups (*p* ≥ 0.29), while repeated measures analysis showed a decrease in pain intensity and an increase in DD symptoms just after caloric or centrifuge stimulation, but not after TOAE stimulation (Table 2). In all participants, no differences were observed between right and left stimulation for any type of stimuli, or between warm caloric and cold caloric stimuli (Table 3).

The pain intensity, based on the VAS, and the frequency of accompanying sensations reported by all the participants six days before any stimulation, and before and after each stimulation, are shown in Figure 1 and Figure 2, respectively. The effects of vestibular stimulation on pain intensity persisted during the following days, while no evident difference was observed after cochlear stimulation. The associated sensations of “tingling”, “pins and needles”, and “itching” decreased after caloric stimulation (t values ranging from 2.44 to 3.80, *p* < 0.02), “tingling” and “pins and needles” decreased after centrifuge stimulation (t values of 3.72 and 2.23, *p* < 0.03), and just “numbness” increased after TOAE stimulation (t value of 2.96, *p* = 0.003).

Individual responses showed that 72 (58.0%, 95% C.I. 49.3–66.6%) participants reported clinically significant pain decrease (>12 mm by VAS) after caloric and/or centrifuge stimulation, while 17 (14%, 95% C.I. 7.9–20.0%) participants reported pain decrease after TOAE stimulation. Among the 72 participants reporting pain decrease after vestibular stimulation, 14 (19%, 95% C.I. 9.9–28.0%) participants reported pain decrease after both caloric and centrifuge stimulation, while none of them reported pain decrease after TOAE stimulation; 34 (47.0%, 95% C.I. 35.4–58.3%) participants reported pain decrease only after caloric stimulation, but 4 (12.0%, 95% C.I. 1.0–22.9%) of them also reported pain decrease after TOAE stimulation, and 1 (3% 95% C.I. 0–7.8%) reported pain increase after TOEA stimulation.; 24 (33.0%, 95% C.I. 22.1–43.8%) participants reported pain decrease only after centrifuge stimulation, but 2 (8%, 95% C.I. −2.8–18.8%) of them also reported pain decrease after TOAE stimulation.

In total, 48 (38%, 95% C.I. 29.4–46.5%) participants reported pain decrease after caloric stimulation, of whom 26 were exposed to right stimuli and 22 to left stimuli, 27 participants were exposed to warm stimuli and 21 were exposed to cold stimuli. In 21 participants, this was the first stimulus; in 7 this was the second stimulus; and in 20, this was the third stimulus. The difference from baseline (before caloric stimulation) persisted for one day in 32 (67%, 95% C.I. 53.7–80.3%) participants, for two days in 27 (56%, 95% C.I. 41.9–72.0%) participants, and for three days in 19 (40%, 95% C.I. 26.1–53.8%) participants.

In total, 38 (30.0%, 95% C.I. 21.9–38.0%) participants reported pain decrease after centrifuge stimulation, of whom 21 participants were exposed to right stimulus and 17 to left stimulus. In 19 participants, this was the first stimulus; in 13, this was the second stimulus; and in 6, this was the third stimulus. The difference from baseline (before centrifuge stimulation) persisted for two days in 12 (32%, 95% C.I. 17.1–46.8%) participants and for three days in 11 (29%, 95% C.I. 14.5–43.4%) participants.

In total, 17 (14%, 95% C.I. 7.9–20.2%) participants reported pain decrease after TOAE stimulation, of whom 9 participants were exposed to right stimulus and eight to left stimulus. In 12 participants, this was the first stimulus, in 2 participants, this was the second stimulus; and in 3 participants, this was the third stimulus. The difference from baseline (before TOAE stimulation) persisted for two days in 11 (65%, 95% C.I. 42.3–87.6%) participants, and for three days in 9 (53%, 95% C.I. 29.2–76.7%) participants.

A total of 47 (38.0%, 95% C.I. 29.4–46.5%) participants reported no clinically significant pain decrease (>12 mm by VAS) after caloric/centrifuge stimulation, including 5 participants with no pain at the moment of stimulation (2 caloric/2 centrifuge/1 centrifuge and TOAE). However, among the 47 participants with no pain decrease after caloric/centrifuge stimulation, 9 (19%, 95% C.I. 7.7–30.2%) reported pain decrease after TOAE stimulation.

Six (5%, 95% C.I. 1.1–8.8%) participants reported pain increase (>12mm by VAS) after caloric or centrifuge stimulation, including five after caloric stimulation and one after centrifuge stimulation. None of them reported a significant pain difference after the other two types of stimuli. Another three (2%, 95% C.I. −0.45–4.4%) participants reported pain increase after TOAE stimulation. At the beginning of the study, these nine participants reported moderate pain (VAS from 4.0 to 6.3) and were 3 to 6 months post-amputation. Two had an anxiety HADS score > 8, one had a depression HADS score > 8, and all but one had a Dissociative Experiences Scale score > 8.

### 3.2. Bivariate Analysis on Depersonalization/Derealization Symptoms

The total score on DD symptoms before and after each type of stimulation is described in Table 2. In the group of 125 participants, both caloric and centrifuge stimulation provoked an increase in the DD symptom score, while TOAEs provoked minimal variation (Table 2); no differences were observed between right and left stimulation, or between warm caloric and cold caloric stimulation (Table 3). None of the participants reported any discomfort related to mental symptoms.

Linear correlations were observed between the difference before and after stimulation in pain intensity and the difference in the total score for DD symptoms. However, the correlation was low for caloric stimulation (r = 0.28, *p* = 0.001), and high for both centrifuge stimulation and (r = −0.97, *p* < 0.00001) and TOAE stimulation (r = −0.93, *p* < 0.00001). The reporting frequency of each DD symptom before any stimulation is described in Table 4. About half of the participants reported “Body feels strange or different in some way”, “Numbing of emotions”, and” Difficulty focusing attention”. Other symptoms were reported by less than half of the participants. The reporting frequency of the five most frequent symptoms at the time of inclusion in the study, both at baseline and after each type of stimulation, is shown in Figure 3. Except for the symptom “Body feels strange or different in some way”, which increased after caloric stimulation, the others were similar after either caloric or centrifuge stimulation, with a decreasing trend after TOAEs.

According to the reported pain decrease after each type of stimulus used, comparisons on the total score of DD symptoms, which includes both frequency and severity of symptoms, between participants reporting pain decrease and those reporting no difference after each stimulus showed that those who reported pain decrease after caloric stimulation had higher DD symptoms total scores than those who reported no difference after the stimulation, both immediately before and after they were exposed to the stimuli (t values of 2.02 to 3.2, d.f. 118, *p* ≤ 0.04). Additionally, those who reported pain decrease after either centrifuge or TOAE had similar DD symptoms scores compared to those who reported no difference after the stimulation (*p* ≥ 0.05).

A comparison of the reporting frequency of each DD symptom showed that both caloric and centrifuge stimulation provoked variable changes in the reporting frequency of DD symptoms, which was not evident after TOAE stimulation (Figure 4). However, participants who reported pain decrease after caloric stimulation reported some DD symptoms more frequently than those who reported no pain difference after the stimulation (t = 2.65–3.36, d.f. 118, *p* ≤ 0.008), including “Surroundings seem strange and unreal”, “Time seems to pass very slowly”, “Feel like you’ve been here before (déjà vu)”, “Feel as though in a dream”, “Numbing of emotions”, “Thoughts seem blurred”, “Events seem to happen in slow motion”, “Your emotions seem disconnected from yourself”, “Feeling of not being in control of self”, and “Feel as though your personality is different”. Meanwhile, differences were minimal after TOAE stimulation (Figure 4), and the participants who reported pain decrease after TOAE reported DD symptoms less frequently than those who reported no pain difference after the stimulation (Figure 4).

### 3.3. Repeated Measures Multivariate Analysis of Pain Intensity

The repeated measures analysis results are described in Table 5. The difference (before and after stimulation) in pain intensity was related to the total score on the HADS, the report of DD symptoms “Body feels strange or different in some way” and “The distinction between close and distant is blurred” before any stimulation, and the report of the associated sensation of “Itching” in the DN4 questionnaire.

However, these variables only contributed to about 15% of the variance (Table 5), with no influence from age, sex, time elapsed since the amputation, or pain intensity at the time of inclusion in the study.

The univariate results show that the report of “Body feels strange or different in some way” only contributed to the difference before-after caloric stimulation, with an increase after the stimuli only in participants with pain decrease (Figure 4). The report of “The distinction between close and distant is blurred” and the total score on the HADS contributed to the difference observed before-after centrifuge and TOAE stimulation, and the report of “Itching” contributed to the difference before-after TOAE stimulation.

The difference in the repeated measures of the pain response to each type of stimulus was related to the reporting frequency of “Body feels strange or different in some way” and the sequence of the stimuli. The largest difference was observed after caloric stimulation among the three types of stimuli, but this effect was minimal when caloric stimulation was the second stimulus after centrifuge stimulation (number 3 in Figure 5). However, in that sequence, among the ten participants with significant pain decrease after centrifuge stimulation, seven reported a persistent pain decrease (similar rating) after centrifuge stimulation. Of note, the pain difference observed after centrifuge and TOAE stimulation was mainly evident when they were applied as the first stimuli (Figure 5).

### 3.4. Repeated Measures Multivariate Analysis of Depersonalization/Derealization Symptoms

The repeated measures analysis results are described in Table 6. The difference (before and after stimulation) in the total score for DD symptoms was related to age, the total score for anxiety/depression by the HADS, and the total score on the Dissociative Experiences Scale, with a borderline result for the sequence of the stimuli, and no influence from sex, time elapsed since the amputation, or pain intensity.

The univariate results show that the HADS total score contributed to the pain response across all types of stimuli. The age and the sequence of the stimuli contributed to the response to caloric stimulation, while the total score on the Dissociative Experiences Scale contributed to the response to centrifuge and to the TOAE stimulation. The differences among the responses to each type of stimuli were also related to age. However, these variables contributed to only about 13% of the variance (Table 6).

## 4. Discussion

This study was aimed at assessing and comparing the effects of caloric semicircular canal stimulation (right or left and warm or cold), centrifuge utricle stimuli (right/left) and cochlear stimuli by TOAE (right/left) on the intensity of pain and DD symptoms reported by adults with phantom pain after ≥3 months since traumatic amputation of hand-finger(s). After vestibular stimuli, pain decrease was reported by more than one-third of the participants, which lasted at least one day in two-thirds of them. Meanwhile, less than one-sixth of the participants reported pain decrease after cochlear stimulation. There were no differences according to which side any of the stimuli were applied or the temperature used in caloric stimulation, but the stimuli sequence had an effect. The effects of the centrifuge and TOAE stimulation were related to the first-time exposure to any stimuli and to the report of anxiety/depression symptoms and a DD symptom related to the environment. After caloric stimulation, pain decrease was independent from the sequence of the stimuli and it was related to the report of a DD symptom related to the body.

In healthy subjects, caloric stimulation can modulate body representation, tactile sensitivity, pain, emotions, and memory (for review, see [40]), while movement of the head can activate the endorphin system to elevate pain thresholds, which can be higher when nodding the head than when sitting still [41]. In patients with neurological or psychiatric disorders, caloric stimulation can also have temporal effects, including on central pain (for review, see [42]), which may not be dependent on mood changes [19].

The convergence of vestibular and nociceptive pathways may substantiate the vestibular modulation of central pain, particularly in the posterior insular nociceptive network. The activation likelihood estimates in 59 studies reporting pain processing have shown activation in the posterior and mid-anterior part of the insula [43].

Painful thermal stimulation of the hands elicits activity over a large network of brain regions, with laterality-specific and attention-specific activity in the insula [44]. Consistently, intraoperative electrical stimulation of the white matter around the posterior insula elicits sensations of pain [45]. Additionally, vestibular activation results in cortical activation of diverse regions, such as the parietal, temporal, insular, and cingulate cortex [46,47]. In particular, evidence supports that the dorsal posterior insula is a multimodal area that can be activated by somatosensory, painful heat, and caloric vestibular stimuli alike [48], as well as visual stimuli [for review, see [49]. Nonetheless, disturbances in the sense of ownership are associated with lesions in the supramarginal gyrus and disconnections of a fronto-insular-parietal network [50].

Both the anterior and posterior subdivisions of the insula can also monitor the emotional salience of afferent signals and integrate it with the stimulus’s effect on the body state [51], with processing occurring at the pre-attentive level [52]. The thalamus, the insula, and the anterior cingulate cortex are fundamental in the integration of salient information [53]. Of note, imaging studies have demonstrated that the insula also has a role in auditory processing, (for review, see [54]), with the anterior subdivision serving as a causal control hub during multisensory attention [55]. In addition, auditory theta entrainment has been effective in reducing chronic pain using low-frequency binaural stimuli (beats) [for review, see [56]. In this study, TOAEs were used monoaurally at 1.5 to 4 KHz.

In this study, the responses to unilateral utricle stimulation did not ascertain or deny a potential effect on pain of graviceptor stimulation, which has been previously assessed in conjunction with semicircular canal stimulation by galvanic vestibular stimulation [22]. However, the unilateral centrifuge stimulation used in this study elicited responses that were related to similar factors contributing to the pain difference elicited after TOAE stimulation. These included being the first stimulus and anxiety/depression symptoms, while perceptual symptoms differed from those evoked by caloric stimulation. These results suggest that the mild unilateral utricle stimulation provided by the centrifuge may have only potentiated the impact of adjuvant factors on pain perception, instead of a main contribution from the vestibular afferents.

Of note, several factors may influence the effect of any intervention to modulate pain, including the placebo effect, the clinical course of the symptoms, and the individual expectations and coping strategies, among others (for review, see [57]). “Placebo effects are beneficial health outcomes not related to the relatively direct biological effects of an intervention and can be elicited by an agent that, by itself, is inert” [58]. These effects comprise a variety of neurobiological and physiological mechanisms, such as endogenous opioid, endocannabinoid, oxytocin, vasopressin, and dopamine systems (for review, see [57]). Simultaneously, positive and negative expectations from a procedure may engage different brain networks to modulate pain experiences, which result in correlated placebo and nocebo behavioral responses [59]. Furthermore, placebo effects and expectations can explain the effects of nonpharmacological treatments (for review, see [60]).

The unexpected finding that the effect of TOAEs on pain was related to the report of “Itching” is in line with the influence of placebo effects and expectations for this result. Neuropathic itching can occur in neurological diseases associated with injury to nervous tissue, including central pain [61]. Corresponding brain areas are activated in pain and itch processing, whereas nocebo and possibly placebo responses can be induced for itching as well as pain, solely by verbal suggestions that manipulate expectations [62].

In this study, the difference in DD symptoms after any stimulation was related to the report of anxiety/depression symptoms, while dissociative experiences were related only to the difference after centrifuge and TOAE stimulation, suggesting that dissociation may have had a role in the response to these stimuli. The majority of individuals experiencing amputation may continue to experience sensory phenomena related to the missing part of the body. Patients with major hand injuries and persistent pain and pressure sensations may report low levels of health satisfaction, with anxiety and depression symptoms [63].

However, the combination of sensations and perceptions, including pain, is distinct among individuals (for review, see [63]), with no simple relationship with psychophysical measures of sensory function [64], but a variety of psychological consequences [65]. In patients with hand injuries or hand disorders, accident location and symptoms of posttraumatic stress disorder can be the most important factors in resuming work [66]. The 10-year follow-up of 83 patients after hand injury showed that one week after the injury, one-quarter of the patients had psychiatric symptoms, predominantly affective symptoms; one year after the injury 10% had persistent symptoms, and after 10 years, some patients still had psychiatric symptoms that could be related to symptoms reported since the third month after the injury [67].

Overall, the results show that mild caloric semicircular canal stimulation, either right/left or warm/cold, may have an influence on phantom pain and symptoms of unreality in some patients. The results are consistent with the hypothesis that an update of the immediate experience of the body, caused by the sensory mismatch induced by asymmetrical vestibular stimulation, may decrease the intensity of phantom limb pain [21].

The pain decrease experienced after utricle stimulation by centrifuge and cochlear stimulation by TOAEs may include placebo effects that could be linked to anxiety/depression symptoms and to individual expectations, which were not particularly assessed in this study. The individual profiles of those who could benefit from pain modulation by any of the types of stimulation included in this study cannot be ascertained from the results. However, the findings support that, during rehabilitation, attention should be paid to both pain and the development of psychiatric symptoms.

The main limitation of this study was a deficient assessment of other psychological components and individual expectations that could modify the responses to any stimuli. The results suggest that these factors could be relevant for both the response to caloric vestibular stimulation and for the placebo/nocebo effects of any stimulation. Even if no participant complained about mental symptoms evoked by the stimuli, future studies may consider including follow-up on mental symptoms and formal psychiatric assessment, as required.

The stimulation effects on pain intensity decreased during the following days, but the period between the stimulation and the follow-up time after completing the protocol were not long enough to assess carryover effects and final responses. The interval period was chosen according to the results from a previous study in adults with diabetes and supracondylar amputation [21]. Therefore, future studies may consider more prolonged intervals and follow-up, according to the characteristics of the participants.

Although pain modulation by various kinds of vestibular stimulation is supported by several studies [17,18,19,20,21,22], their translation to clinical scenarios, such as those encountered in central chronic pain and functional disorders, requires consideration of the target disease, comorbidities, and idiosyncrasies to select patients and design stimuli protocols.

The cross-over design of this study allowed for comparison of a variety of stimuli within the same individual but required careful selection of the stimuli to prevent abandonment of the protocol. To support potential translations to clinical practice, future studies may include sham stimuli, as well as repeated stimuli in sequential sessions.

The results of this study, from a homogenous cohort without complicating diseases, provide useful information highlighting that mild caloric vestibular stimulation, in either the right or left side and at a warm or cold temperature, can modulate phantom pain after amputation of finger(s) in patients with altered bodily sensations. However, individual differences and expectations may have an influence on the response to these and similar types of stimulation.

## Figures and Tables

**Figure 1 biomedicines-13-01601-f001:**
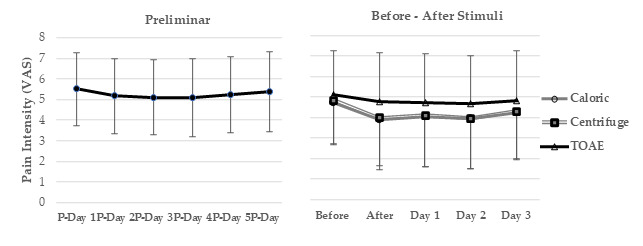
Mean and standard deviation of the mean of pain intensity six days before any stimulus, and before and after caloric/centrifuge/transient evoked otoacoustic emissions (TOAEs) stimulation in the 125 participants.

**Figure 2 biomedicines-13-01601-f002:**
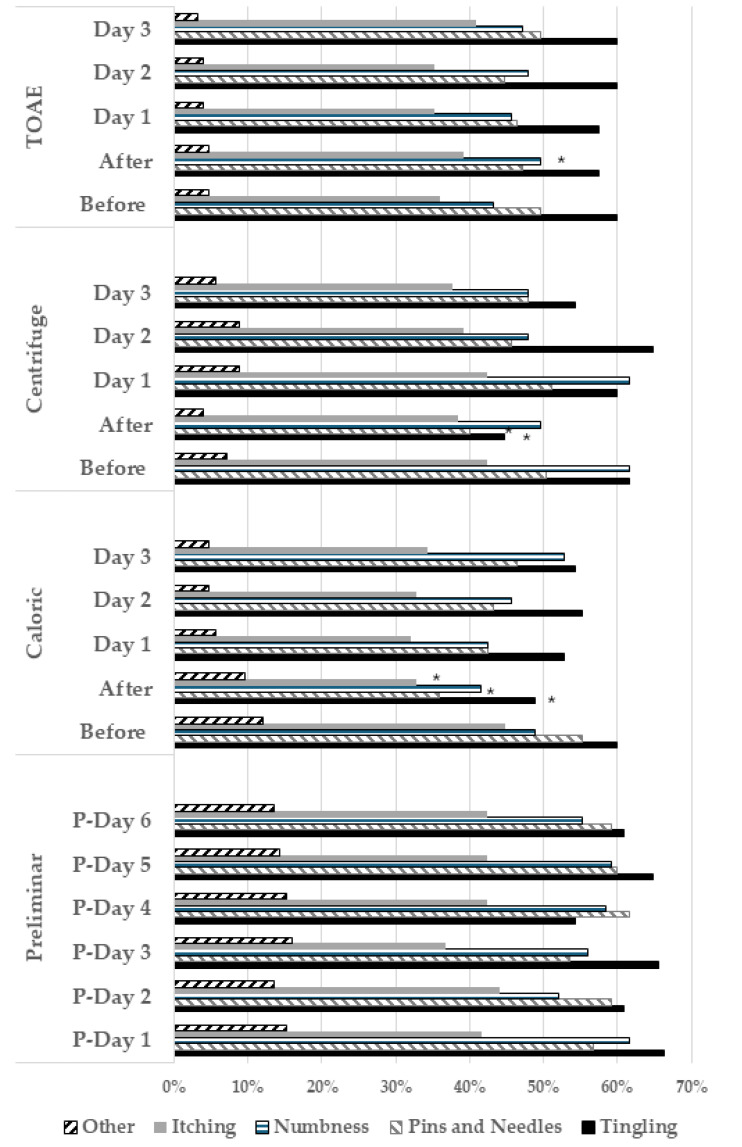
Frequency of accompanying sensations every day for six days before any stimulus and before and after caloric/centrifuge/transient evoked otoacoustic emissions (TOAEs) stimulation in the 125 participants. Statistical differences are highlighted using *.

**Figure 3 biomedicines-13-01601-f003:**
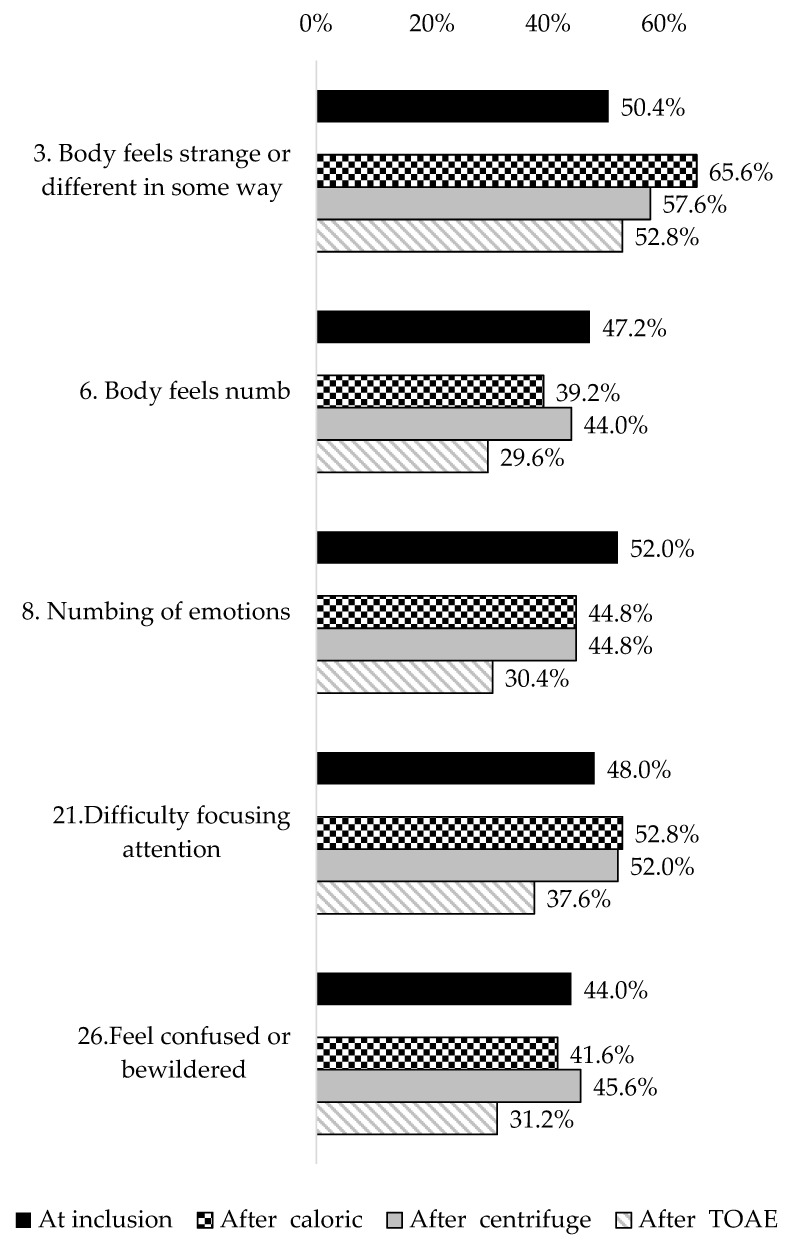
Frequency of the five most frequent symptoms at the time of inclusion in the study before any stimulation and after each type of stimulation of 125 participants with phantom pain after traumatic hand-finger(s) amputation.

**Figure 4 biomedicines-13-01601-f004:**
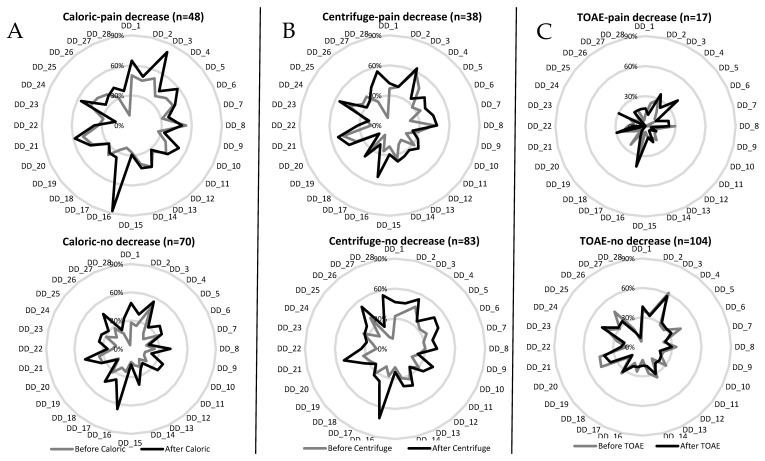
Reporting frequency of each depersonalization/derealization symptom before and after the three types of stimuli were applied in participants who reported either pain decrease or no pain decrease. Note the differences between the subgroups (pain decrease/no decrease) observed after caloric stimulation (**A**), centrifuge stimulation (**B**) versus cochlear stimulation (TOAE) (**C**).

**Figure 5 biomedicines-13-01601-f005:**
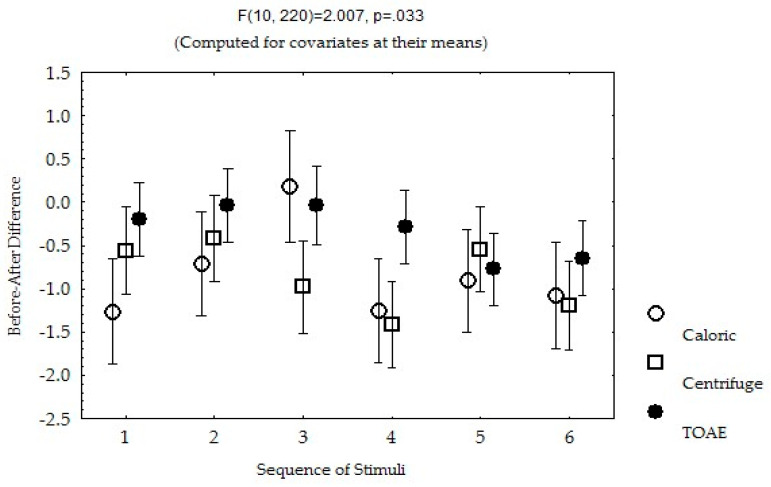
Mean and standard error of the mean difference before and after stimulation, according to the sequence of the stimuli: (1) caloric/centrifuge/TOAE; (2) caloric/TOAE/centrifuge; (3) centrifuge/caloric/TOAE; (4) centrifuge/TOAE/caloric; (5) TOAE/caloric/centrifuge; (6) TOAE/centrifuge/caloric.

**Table 1 biomedicines-13-01601-t001:** Pain characteristics and common mental symptoms before the 125 participants were exposed to any stimulation. S.D. = standard deviation, TOAE = transient otoacoustic emissions.

Variable	All	First Stimulus Subgroups		
	(*n* = 125)	Caloric (*n* = 42)	Centrifuge (*n* = 41)	TOAE (*n* = 42)
	Mean ± S.D.	Mean ± S.D.	Mean ± S.D.	Mean ± S.D.
Years of age	38.2 ± 8.1	40.1 ± 9.0	37.3 ± 7.6	37.3 ± 7.4
Body mass index	27.7 ± 4.1	27.3 ± 3.7	27.7 ± 4.2	28.0 ± 4.5
Visual analog scale of pain	5.4 ± 1.7	5.3 ± 1.6	5.4 ± 1.5	5.8 ± 1.9
PAIN CHARACTERISTICS (DN4)	Percentage (95% C.I.)	Percentage (95% C.I.)	Percentage (95% C.I.)	Percentage (95% C.I.)
Electric shocks	44.8 (35.9–53.6)	38.0 (22.7–53.4)	60.9 (45.3–76.5)	35.7 (20.6–50.8)
Painful cold	55.2 (46.3–64.0)	50.0 (34.2–65.7)	65.8 (50.7–81.0)	50.0 (34.7–65.7)
Burning	57.6 (48.5–66.6)	59.5 (44.0–75.0)	48.7 (31.3–66.2)	34.2 (49.1–79.3)
Associated symptoms				
Tingling	59.2 (50.4–67.9)	61.9 (46.5–77.2)	65.8 (50.7–81.0)	50 3 (4.2–65.7)
Pins and needles	56.0 (47.1–64.8)	66.6 (51.7–81.5)	48.7 (32.8–64.7)	52.3 (36.6–68.1)
Numbness	60.8 (52.1–69.4)	57.1 (41.5–72.7)	58.5 (42.7–74.2)	66.6 (51.7–81.5)
Itching	45.6 (36.7–54.4)	45.2 (29.5–60.9)	43.9 (28.0–59.7)	47.6 (31.8–63.3)
	Median (Quartiles 1–3)	Median (Quartiles 1–3)	Median (Quartiles 1–3)	Median (Quartiles 1–3)
Months elapsed since amputation	4 (4–6)	5 (3–6)	4 (4–5)	4 (3–5)
LATTINEN INDEX total score	7 (5–8)	7 (5–10)	7 (6–8)	7 (6–8)
Pain intensity	2 (1–3)	2 (1–2)	2 (1–2)	2 (1–3)
Pain frequency	2 (2–3)	2 (2–3)	2 (1–3)	2 (2–3)
Need for medication	1 (0–1)	1 (0–1)	1 (0–1)	1 (0–1)
Handicap	1 (0–2)	1 (0–2)	1 (0–2)	1 (0–2)
Sleep time	1 (0–2)	1 (0–2)	1 (0–2)	1 (0–2)
MENTAL SYMPTOMS				
Anxiety and depression	14 (10–17)	14.5 (10–18)	14 (10–17)	13.5 (10–18)
Anxiety	7 (5–10)	8 (4–10)	7 (5–9)	8.5 (5–11)
Depression	6 (4–8)	6 (4–8)	7 (4–9)	5.5 (3–8)
Dissociative experiences	10.0 (6.4–16.7)	11.0 (5.3–17.8)	10.3 (7.5–16.4)	9.1 (6.4–16.7)
Depersonalization/derealization	7 (3–17)	8 (5–21)	5 (2–12)	7 (3–16)

**Table 2 biomedicines-13-01601-t002:** Mean and standard deviation of the mean of pain intensity and depersonalization/derealization score. Comparisons were performed by repeated measures analysis of variance and Tukey’s honest significance difference (T.H.S.D.) test. S.D. = standard deviation; d.f. = degrees of freedom.

Variable	Mean ± S.D.	Mean ± S.D.	*p* (F, d.f.)	T.H.S.D. *p*
Pain Intensity	Before	After	<0.00001 (17.6, 5)	
Caloric	4.7 ± 2.0	3.8 ± 2.3		0.00002
Centrifuge	4.8 ± 2.1	3.9 ± 2.3		0.00002
Transient Otoacoustic Emissions	5.0 ± 2.1	4.7 ± 2.3		0.30
Depersonalization/Derealization Score	Before	After	<0.00001 (17.9, 5)	
Caloric	9.5 ± 10.9	13.2 ± 9.8		0.00002
Centrifuge	9.2 ± 10.2	13.2 ± 8.8		0.00002
Transient Otoacoustic Emissions	9.1 ± 9.8	8.2 ± 7.7		0.84

**Table 3 biomedicines-13-01601-t003:** Mean and standard deviation of the mean of the pain intensity and depersonalization/derealization (DD) score, before and after stimulation. Comparisons were performed by analysis of variance (ANOVA) for caloric stimuli and *t* tests for centrifuge and transient otoacoustic emissions (TOAEs) stimulation. S.D. = standard deviation; d.f. = degrees of freedom.

	Caloric					Centrifuge			TOAE		
	Right 30 °C (*n* = 32)	Right 44 °C (*n* = 32)	Left 30 °C (*n* = 31)	Left 44 °C (*n* = 30)	ANOVA	Right (*n* = 60)	Left (*n* = 65)	*t* test	Right (*n* = 64)	Left (*n* = 61)	*t* test
Pain intensity	Mean ± S.D.	Mean ± S.D.	Mean ± S.D.	Mean ± S.D.	*p* (F, d.f.)	Mean ± S.D.	Mean ± S.D.	*p* (t, d.f.)	Mean ± S.D.	Mean ± S.D.	*p* (t, d.f.)
Before	4.8 ± 1.9	4.8 ± 1.6	4.6 ± 2.0	4.5 ± 2.4	0.95 (0.10; 3, 121)	4.7 ± 2.2	4.8 ± 2.0	0.71 (0.36, 123)	4.9 ± 2.1	5.2 ± 2.2	0.3 2 (0.99, 123)
After	4.0 ± 2.6	3.7 ± 1.8	3.6 ± 2.4	4.0 ± 2.7	0.84 (0.27; 3, 121)	3.9 ± 2.6	4.0 ± 2.0	0.75 (0.31, 123)	4.6 ± 2.4	4.8 ± 2.2	0.51 (0.65, 123)
Difference	−0.7 ± 1.7	−1.0 ± 1.6	−1.0 ± 1.1	−0.5 ± 1.3	0.40 (0.98; 3, 121)	−0.8 ± 1.3	−0.8 ± 1.1	0.96 (0.39, 123)	−0.3 ± 0.9	−0.36 ± 1.1	0.85 (0.18, 123)
DD symptoms	Mean ± S.D.	Mean ± S.D.	Mean ± S.D.	Mean ± S.D.	p (F, d.f.)	Mean ± S.D.	Mean ± S.D.	p (t, d.f.)	Mean ± S.D.	Mean ± S.D.	p (t, d.f.)
Before	8.9 ± 10.6	10.6 ± 12.0	8.8 ± 10.7	9.8 ± 10.9	0.90 (0.19; 3, 121)	10.0 ± 11.2	8.5 ± 9.0	0.41 (0.81, 123)	9.1 ± 8.2	9.0 ± 11.3	0.92 (0.10, 121)
After	12.5 ± 7.4	13.5 ± 11.9	13.7 ± 9.0	13.2 ± 10.8	0.96 (0.09; 3, 121)	13.7 ± 8.7	12.7 ± 9.0	0.51 (0.65, 123)	7.7 ± 6.2	8.6 ± 9.0	0.50 (0.67, 121)
Difference	3.6 ± 6.6	2.9 ± 9.4	4.8 ± 5.7	3.6 ± 6.9	0.72 (0.43; 3, 121)	3.7 ± 8.1	4.2 ± 5.6	0.72 (0.35, 123)	−1.4 ± 4.5	−0.3 ± 6.2	0.25 (1.1, 121)

**Table 4 biomedicines-13-01601-t004:** Frequency of depersonalization/derealization symptoms reported at the time of inclusion in the study by 125 participants with phantom pain after traumatic hand-finger(s) amputation.

Symptom	Percentage (95% C.I.)	*n*
1. Surroundings seem strange and unreal	39.2% (30.6–47.7)	49
2. Time seems to pass very slowly	36.0% (27.5–44.4)	45
3. Body feels strange or different in some way	50.4% (41.6–59.1)	63
4. Feel like you’ve been here before (déjà vu)	38.4% (29.8–46.9)	48
5. Feel as though in a dream	35.2% (26.8–43.5)	44
6. Body feels numb	47.2% (38.4–55.9)	59
7. Feeling of detachment or separation from surroundings	28.0% (20.1–35.8)	35
8. Numbing of emotions	52.0% (43.2–60.7)	65
9. People and objects seem far away	23.2% (15.8–30.6)	29
10. Feeling detached or separated from your body	22.4% (15.0–29.7)	28
11. Thoughts seem blurred	38.4% (29.8–46.9)	48
12. Events seem to happen in slow motion	23.2% (15.8–30.6)	29
13. Your emotions seem disconnected from yourself	31.2% (23.0–39.3)	39
14. Feeling of not being in control of self	30.4% (22.3–38.4)	38
15. People appear strange or unreal	25.6% (17.9–33.2)	32
16. Dizziness	36.8% (28.3–45.2)	46
17. Surroundings appear covered with a haze	14.4% (8.2–20.5)	18
18. Vision is dulled	32.8% (24.5–41.0)	41
19. Feel as if walking on shifting ground	17.6% (10.9–24.2)	22
20. Difficulty understanding what others say to you	29.6% (21.6–37.6)	37
21. Difficulty focusing attention	48.0% (39.2–56.7)	60
22. Feel as though in a trance	26.4% (18.6–34.1)	33
23. The distinction between close and distant is blurred	30.4% (22.3–38.4)	38
24. Difficulty concentrating	42.4% (33.7–51.0)	53
25. Feel as though your personality is different	34.4% (26.0–42.7)	43
26. Feel confused or bewildered	44.0% (35.3–52.7)	55
27. Feel isolated from the world	26.4% (18.6–34.1)	33
28. Feel “spacy” or “spaced out”	8.0% (3.2–12.7)	10

**Table 5 biomedicines-13-01601-t005:** Results of the repeated measures multivariate analysis of covariance on the difference (before and after stimulation) in pain intensity for each type of stimulation in all participants. Statistical significance is highlighted using *. DD = Depersonalization/Derealization Inventory, DN4 = *Douleur Neuropathique* 4 Questionnaire d.f. = degrees of freedom.

	Repeated Measures	Caloric	Centrifuge	TOAE
Variables		Adjusted R^2^ (F Value, *p*) 0.15 (2.59, 0.002)	Adjusted R^2^ (F Value, *p*) 0.15 (2.61, 0.002)	Adjusted R^2^ (F Value, *p*) 0.17 (2.93, 0.0008)
	*p* (F Value, d.f.)	*p* (F Value, d.f.)	*p* (F Value, d.f.)	*p* (F Value, d.f.)
Intercept	0.028 (4.92, 1) *	0.044 (4.12) *	0.39 (0.73)	0.60 (0.26)
Anxiety/depression HADS total score	0.019 (5.63, 1) *	0.68 (0.16)	0.012 (6.49) *	0.024 (5.23) *
First DD_3: “Body feels strange or different in some way”	0.026 (5.08, 1) *	0.005 (7.86) *	0.11 (2.46)	0.17 (1.84)
First DD_23: “The distinction between close and distant is blurred”	0.007 (7.29, 1) *	0.71 (0.13)	0.018 (5.69) *	0.002 (9.45) *
Sequence of the stimuli	0.001 (4.10, 5) *	0.016 (2.91) *	0.029 (2.59) *	0.06 (2.11)
First DN4_7: Itching	0.0002 (14.63, 1) *	0.07 (3.33)	0.11 (2.56)	0.002 (9.72) *
Sequence of stimuli * and DN4_7: Itching	0.42 (1.25, 5)	0.33 (1.15)	0.47 (0.91)	0.46 (0.93)
R1 Repeated measures (R1)	0.36 (1.01, 2)	-	-	-
R1 and HADS score	0.10 (2.32, 2)	-	-	-
R1 and DD1_3	0.01 (4.67, 2) *	-	-	-
R1 and DD1_23	0.07 (2.00, 2)	-	-	-
R1 and sequence of stimuli	0.03 (2.59, 10) *	-	-	-
R1 and DN1_7	0.78 (0.24, 2)	-	-	-
Sequence of stimuli and DN4_7	0.40 (1.04, 10)	-	-	-

**Table 6 biomedicines-13-01601-t006:** Results of the repeated measures multivariate analysis of covariance of the difference (before and after stimulation) in the total score on the Depersonalization/Derealization Inventory, for each type of stimuli in all participants. Statistical significance is highlighted using *. d.f. degrees of freedom.

	Repeated Measures	Caloric	Centrifuge	TOAE
Variables		Adjusted R^2^ (F Value, *p*) 0.13 (3.48, 0.001)	Adjusted R^2^ (F Value, *p*) 0.13 (3.52, 0.001))	Adjusted R^2^ (F Value, *p*) 0.15 (3.78, 0.0005))
	*p* (F Value, d.f.)	*p* (F Value, d.f.)	*p* (F Value, d.f.)	*p* (F Value, d.f.)
Intercept	0.0000001 (31.24, 1) *	0.00001 (19.78)	0.00002 (19.30)	0.021 (5.43)
Age	0.006 (7.62, 1) *	0.015 (6.01) *	0.07 (3.21)	0.19 (1.73)
Anxiety/depression HADS Score	0.0003 (13.32, 1) *	0.015 (6.07) *	0.004 (8.25) *	0.039 (4.33) *
Dissociative Experiences Score	0.0001 (14.97, 5) *	0.21 (1.57)	0.004 (8.46) *	0.00006 (17.01) *
Sequence of stimuli	0.054 (2.24, 2)	0.022 (2.73) *	0.15 (1.64)	0.41 (1.01)
Repeated measures (R1)				
R1 * Age	0.038 (3.29, 2) *	-	-	-
R1 * HADS Score	0.48 (0.72, 2)	-	-	-
R1 * Dissociative Experiences Score	0.54 (0.61, 2)	-	-	-
R1 * Sequence of stimuli	0.20 (1.60, 10)	-	-	-

## Data Availability

All data generated or analyzed during this study are included in this article. Further enquiries can be directed to the corresponding author.

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
