# Peer review of "Vestibular Versus Cochlear Stimulation on the Relief of Phantom Pain After Traumatic Finger Amputation"

_biomedicines, 2025, doi:10.3390/biomedicines13071601_

Round 1
Reviewer 1 Report
Comments and Suggestions for Authors
This manuscript presents a randomized crossover study evaluating the effects of vestibular stimuli (caloric and centrifuge) versus cochlear stimuli (TOAE) on phantom limb pain and depersonalization/derealization (DD) symptoms in adults with traumatic hand-finger amputations. The authors report that caloric vestibular stimuli led to pain relief in a significant subset of participants, while cochlear stimuli were less effective. Notably, the study links these effects with accompanying dissociative symptoms and psychological profiles, suggesting potential clinical implications for vestibular neuromodulation in pain management.
Major Comments and Suggestions
- Expand the discussion on the neurobiological mechanisms by which vestibular stimuli may influence pain perception. Include references to relevant insular cortex, multisensory integration, and body ownership pathways to help bridge the observed clinical effects with theoretical underpinnings.
- Since treatment order appeared to influence outcomes, explain how randomization and washout periods were intended to mitigate bias. Consider discussing whether participants’ expectations or prior exposure to stimuli might have played a role in the results.
- As some participants reported increases in depersonalization/derealization (DD) symptoms after vestibular stimulation, clarify:
- Whether these effects were short-lived or persistent.
- If participants found them disturbing or neutral.
- Whether they should be seen as therapeutic side effects or potential concerns.
- Simplify or reorganize complex figures to make key trends clearer. For example:
- Combine similar symptoms in grouped bar charts or a heatmap.
- Highlight statistically significant changes with markers or labels.
- Emphasize the need for follow-up assessments to evaluate whether the observed pain relief is sustained or if effects wane over time. Recommend future trials with multiple sessions and longer monitoring windows.
- The auditory (TOAE) condition may still stimulate neural activity. For more robust conclusions, suggest future studies incorporate a sham vestibular stimulus (e.g., room-temperature irrigation without caloric effect) to isolate vestibular-specific outcomes.
- Ensure that all acronyms are defined on first use and technical terms are clearly explained for a broader clinical audience. Consider revising for smoother phrasing throughout the manuscript.
- Briefly speculate how these findings could inform non-invasive neuromodulation therapies for other chronic pain conditions or psychiatric symptoms involving altered bodily self-awareness (e.g., CRPS, fibromyalgia).
- Mechanistic Explanation Needs Expansion:
- The proposed link between vestibular modulation and posterior insular nociceptive networks is promising but underexplored.
- Suggestion: Expand the discussion on the neural pathways mediating vestibular influence on pain (e.g., vestibulo-insular integration, multisensory body representation, interoception).
- Clarify Sequence Effects and Confounders:
- The results show that centrifuge and TOAE stimuli effects were stronger when administered first. This raises concerns about expectation bias or learning effects.
- Suggestion: Include a discussion on potential carryover effects and how randomization and washout periods (3 days) mitigate this.
- Differentiate Between Pain Relief and Dissociative Symptoms:
- The observed correlation between pain relief and increased DD symptoms (particularly after caloric stimuli) may raise safety or ethical concerns.
- Suggestion: Clarify whether the dissociative symptoms were transient and perceived as distressing or neutral/beneficial by participants.
- Lack of Placebo Control:
- While TOAE was used as a control, its auditory stimulation could still activate neural pathways unrelated to vestibular modulation.
- Suggestion: Consider including a sham vestibular stimulation condition in future studies or discuss the limitation more directly.
- Long-Term Effects Missing:
- The follow-up was limited to three days post-stimulus. No information is given on longer-term persistence or habituation.
- Suggestion: Propose future longitudinal studies to explore the clinical applicability of repeated vestibular interventions.
- Figures Are Overloaded:
- Figures 3 and 4 include numerous symptoms plotted, which may overwhelm readers.
- Suggestion: Summarize key symptom patterns or consolidate into heatmaps or grouped bar plots to improve interpretability.
- Some abbreviations (e.g., DD, TOAE) are used early without definition; recommend standardizing first-use explanations.
- The description of stimuli devices (e.g., "I-Portal NOTC") should be more reader-friendly for broader audiences.
- Some English phrasing is awkward (e.g., “Body fells strange” → “Body feels strange”). A light language polish is advised.
- Were participants debriefed about the nature of vestibular vs cochlear stimuli post-trial to assess expectation or placebo awareness?
- Could participants distinguish between the stimuli types based on their sensory experience, and might this have influenced outcomes?
- Did any participants experience adverse effects such as vertigo or nausea from vestibular stimuli?
- Do the authors plan to explore repetitive or home-based vestibular stimulation protocols in future research?
Comments on the Quality of English Language
The manuscript is largely readable but includes minor grammatical errors and occasionally awkward phrasing. Some sentences are too long and would benefit from restructuring. Suggest professional proofreading before final submission.
Author Response
This manuscript presents a randomized crossover study evaluating the effects of vestibular stimuli (caloric and centrifuge) versus cochlear stimuli (TOAE) on phantom limb pain and depersonalization/derealization (DD) symptoms in adults with traumatic hand-finger amputations. The authors report that caloric vestibular stimuli led to pain relief in a significant subset of participants, while cochlear stimuli were less effective. Notably, the study links these effects with accompanying dissociative symptoms and psychological profiles, suggesting potential clinical implications for vestibular neuromodulation in pain management.
- We thank the reviewer for the insightful comment.
Expand the discussion on the neurobiological mechanisms by which vestibular stimuli may influence pain perception. Include references to relevant insular cortex, multisensory integration, and body ownership pathways to help bridge the observed clinical effects with theoretical underpinnings.
- The introduction (lines 78-85) and discussion (lines 430-439) were expanded according to the recommendation and 23 more references were included .
Since treatment order appeared to influence outcomes, explain how randomization and washout periods were intended to mitigate bias. Consider discussing whether participants’ expectations or prior exposure to stimuli might have played a role in the results.
- The limitation paragraphs were edited (lines 512-517) and the possible influence of expectations is discussed (lines 459-468), while a reference is provided for review on the topic.
As some participants reported increases in depersonalization/derealization (DD) symptoms after vestibular stimulation, clarify:
Whether these effects were short-lived or persistent.
- At the end of the Discussion section a recommendation of follow-up on mental symptoms was incorporated (lines 509-511).
If participants found them disturbing or neutral. Whether they should be seen as therapeutic side effects or potential concerns.
- Although no formal assessment was performed, none of them reported discomfort (lines 303-304).
Simplify or reorganize complex figures to make key trends clearer. For example:
Combine similar symptoms in grouped bar charts or a heatmap. Highlight statistically significant changes with markers or labels.
- Figures 2,3 and 4 have been modified/amplified for clarity, keeping in mind the main aim of the study ( comparison among stimuli effects). Statistical differences described in the text are now highlighted in Figure 2. Data labels were added to Figure 3. The legend of Figure 4 now highlights the main contrast among stimuli.
Emphasize the need for follow-up assessments to evaluate whether the observed pain relief is sustained or if effects wane over time. Recommend future trials with multiple sessions and longer monitoring windows.
- The emphasis and recommendation were included at the end of the Discussion section (lines 512-517).
The auditory (TOAE) condition may still stimulate neural activity. For more robust conclusions, suggest future studies incorporate a sham vestibular stimulus (e.g., room-temperature irrigation without caloric effect) to isolate vestibular-specific outcomes.
- The Recommendation was included at the end of the Discussion section (lines 524.525), which was expanded to include the auditory stimuli (lines 444-448).
Ensure that all acronyms are defined on first use and technical terms are clearly explained for a broader clinical audience. Consider revising for smoother phrasing throughout the manuscript.
- All acronyms are defined within the text and at the end of the manuscript. The revision was performed.
Briefly speculate how these findings could inform non-invasive neuromodulation therapies for other chronic pain conditions or psychiatric symptoms involving altered bodily self-awareness (e.g., CRPS, fibromyalgia).
- Some comments were included. However, the study was limited and further speculation could be misleading (lines 517-521).
Mechanistic Explanation Needs Expansion:
The proposed link between vestibular modulation and posterior insular nociceptive networks is promising but underexplored. Suggestion: Expand the discussion on the neural pathways mediating vestibular influence on pain (e.g., vestibulo-insular integration, multisensory body representation, interoception).
- Thank you for the insightful comment, the Discussion was expanded (lines 430-439).
Clarify Sequence Effects and Confounders:
The results show that centrifuge and TOAE stimuli effects were stronger when administered first. This raises concerns about expectation bias or learning effects.
- In the Discussion section, emphasis is given on future studies assessing expectations (465-469 and 505-508) .
Suggestion: Include a discussion on potential carryover effects and how randomization and washout periods (3 days) mitigate this.
- The limitations paragraphs were edited accordingly (lines 512-517).
Differentiate Between Pain Relief and Dissociative Symptoms:
The observed correlation between pain relief and increased DD symptoms (particularly after caloric stimuli) may raise safety or ethical concerns. Suggestion: Clarify whether the dissociative symptoms were transient and perceived as distressing or neutral/beneficial by participants.
- The Results section now clarify that none of the participants reported discomfort related to mental symptoms (lines 303-304) and a recommendation was included on the need to follow-up mental symptoms, as well as psychiatric assessment, as required (lines 509-511).
Lack of Placebo Control:
While TOAE was used as a control, its auditory stimulation could still activate neural pathways unrelated to vestibular modulation. Suggestion: Consider including a sham vestibular stimulation condition in future studies or discuss the limitation more directly.
- The recommendation was included (lines 522-525) and the discussion was expanded (lines 542-548).
Long-Term Effects Missing: The follow-up was limited to three days post-stimulus. No information is given on longer-term persistence or habituation. Suggestion: Propose future longitudinal studies to explore the clinical applicability of repeated vestibular interventions.
- The suggestion was included (lines 516-517).
Figures Are Overloaded:
Figures 3 and 4 include numerous symptoms plotted, which may overwhelm readers. Suggestion: Summarize key symptom patterns or consolidate into heatmaps or grouped bar plots to improve interpretability.
- Thank you for the recommendation, Figures 2,3 and 4 have been modified.
Some abbreviations (e.g., DD, TOAE) are used early without definition; recommend standardizing first-use explanations.
- The manuscript was revised to ensure early and repetitive definition of acronyms, including definitions at the end of the manuscript as well.
The description of stimuli devices (e.g., "I-Portal NOTC") should be more reader-friendly for broader audiences.
- That is the name of the equipment; for better comprehension of the stimuli, the description of the centrifuge stimuli was expanded (lines 174-181).
Some English phrasing is awkward (e.g., “Body fells strange” → “Body feels strange”). A light language polish is advised.
- We apologize for the error. The manuscript was revised.
Were participants debriefed about the nature of vestibular vs cochlear stimuli post-trial to assess expectation or placebo awareness?
- No, the protocol did not include these assessments
Could participants distinguish between the stimuli types based on their sensory experience, and might this have influenced outcomes?
- The study did not include any formal assessment on this point.
Did any participants experience adverse effects such as vertigo or nausea from vestibular stimuli?
- All participants reported vertigo during caloric stimuli, this experience was used to ascertain adequate stimuli along with the evidence of nystagmus (lines 186-187), but none of them experienced nausea. All participants were informed about the symptoms that could be elicited by the stimulus, both before they provided their written consent to participate and before vestibular stimulus.
Do the authors plan to explore repetitive or home-based vestibular stimulation protocols in future research?
- The up-to date evidence is yet insufficient to design treatment protocols
Comments on the Quality of English Language
The manuscript is largely readable but includes minor grammatical errors and occasionally awkward phrasing. Some sentences are too long and would benefit from restructuring. Suggest professional proofreading before final submission.
- We thank the reviewer for the comment . The manuscript was revised
Reviewer 2 Report
Comments and Suggestions for Authors
This is a well-designed and carefully executed experimental study addressing a very specific but clinically relevant topic: the modulation of phantom limb pain via vestibular versus cochlear stimulation. The cross-over design, inclusion of multiple stimuli types, and comprehensive assessment of both pain and dissociative symptoms strengthen the conclusions. Nevertheless, the introduction could benefit from a slightly more detailed explanation of the proposed neurophysiological mechanisms underpinning vestibular modulation of pain, incorporating some more recent neuroimaging and neurophysiological literature.
Author Response
This is a well-designed and carefully executed experimental study addressing a very specific but clinically relevant topic: the modulation of phantom limb pain via vestibular versus cochlear stimulation. The cross-over design, inclusion of multiple stimuli types, and comprehensive assessment of both pain and dissociative symptoms strengthen the conclusions. Nevertheless, the introduction could benefit from a slightly more detailed explanation of the proposed neurophysiological mechanisms underpinning vestibular modulation of pain, incorporating some more recent neuroimaging and neurophysiological literature.
- We thank the reviewer for the positive comments and we have edited the Introduction according to the suggestions provided.
Round 2
Reviewer 1 Report
Comments and Suggestions for Authors
Thank you for addressing the comments. It should be good to go.
Comments on the Quality of English LanguageThank you for addressing the comments. It should be good to go.